# The Cellular Abundance of Chemoreceptors, Chemosensory Signaling Proteins, Sensor Histidine Kinases, and Solute Binding Proteins of *Pseudomonas aeruginosa* Provides Insight into Sensory Preferences and Signaling Mechanisms

**DOI:** 10.3390/ijms24021363

**Published:** 2023-01-10

**Authors:** Miguel A. Matilla, Roberta Genova, David Martín-Mora, Sandra Maaβ, Dörte Becher, Tino Krell

**Affiliations:** 1Department of Biotechnology and Environmental Protection, Estación Experimental del Zaidín, Consejo Superior de Investigaciones Científicas, 18008 Granada, Spain; 2Department of Microbial Proteomics, Institute of Microbiology, University of Greifswald, D-17489 Greifswald, Germany

**Keywords:** chemotaxis, chemoreceptor, sensor histidine kinases, solute binding protein, chemosensory pathway, protein abundance, *Pseudomonas aeruginosa*

## Abstract

Chemosensory pathways and two-component systems are important bacterial signal transduction systems. In the human pathogen *Pseudomonas aeruginosa,* these systems control many virulence traits. Previous studies showed that inorganic phosphate (Pi) deficiency induces virulence. We report here the abundance of chemosensory and two-component signaling proteins of *P. aeruginosa* grown in Pi deficient and sufficient media. The cellular abundance of chemoreceptors differed greatly, since a 2400-fold difference between the most and least abundant receptors was observed. For many chemoreceptors, their amount varied with the growth condition. The amount of chemoreceptors did not correlate with the magnitude of chemotaxis to their cognate chemoeffectors. Of the four chemosensory pathways, proteins of the Che chemotaxis pathway were most abundant and showed little variation in different growth conditions. The abundance of chemoreceptors and solute binding proteins indicates a sensing preference for amino acids and polyamines. There was an excess of response regulators over sensor histidine kinases in two-component systems. In contrast, ratios of the response regulators CheY and CheB to the histidine kinase CheA of the Che pathway were all below 1, indicative of different signaling mechanisms. This study will serve as a reference for exploring sensing preferences and signaling mechanisms of other bacteria.

## 1. Introduction

The capacity of bacteria to adapt to changing environmental conditions relies on an array of receptors that sense different signals. Major families of such receptors include sensor histidine kinases; chemoreceptors; adenylate-, diadenylate-, and diguanylate cyclases; cAMP-, c-di-AMP-, and c-di-GMP phosphodiesterases; protein kinases; and phosphatases [1]. Many bacterial strains contain an elevated number of sensor histidine kinases and chemoreceptors [2], which are the sensory proteins of two-component systems and chemosensory pathways, respectively. These regulatory circuits are of central relevance for the regulation of multiple physiological and metabolic processes [3,4].

A canonical two-component system is composed of a sensor histidine kinase and a response regulator. Typically, signal recognition at the sensor histidine kinase modulates its autokinase activity, leading subsequently to changes in the transphosphorylation kinetics to the response regulator. The regulatory output is defined by the ratio of phosphorylated to non-phosphorylated response regulators [4]. The analysis of the response regulator output domains indicates that a large part of two-component systems are involved in transcriptional regulation, whereas other systems are involved in cAMP or c-di-GMP signaling, or bind to RNA [5]. There is only a limited amount of cross-talk between different two-component systems [6], indicating that in most cases the binding of a signal molecule to a sensor histidine kinase modulates the phosphorylation state of a single protein, the cognate response regulator.

This situation is quite different in chemosensory pathways, which correspond to sophisticated versions of two-component systems. Most chemosensory pathways mediate chemotaxis, whereas other pathways regulate second messenger levels or are associated with twitching motility [7]. Many bacteria have multiple chemosensory pathways [2,3], as a mechanism to integrate different environmental signals to control various cellular processes. Signal (e.g., chemoeffector) recognition at chemoreceptors modulates the autokinase activity CheA. Typically, CheA is stimulated by many chemoreceptors that all recognize different environmental stimuli [8,9,10,11]. Perhaps one of the most remarkable examples is that of the cattle gastrointestinal tract isolate *Caryophanon latum* [12], whose genome encodes 90 chemoreceptors that are likely to stimulate its sole chemosensory pathway [2]. The histidine kinase CheA phosphorylates two response regulators, namely CheY, which mediates the signaling output, and CheB, which plays a key role in the adaptation of the system to the present signal concentration [2,9,13]. Therefore, and in marked contrast to two-component systems, multiple chemoreceptors define the phosphorylation state of a single protein, CheY. This implies that the relative abundance of chemoreceptors defines the weight they have in modulating the CheY-P concentration.

This issue is directly related to a major research need in the study of chemotaxis, which consists of delineating the mechanisms by which bacteria define responses in natural habitats that are frequently characterized by complex chemoeffector mixtures. The amount of chemoreceptors may thus be a means to prioritize and bias responses to certain signals. In addition, there are different regulatory processes at the transcriptional and post-transcriptional levels [9] that alter the amount of chemoreceptors and consequently the corresponding response bias.

Information on protein amounts of chemoreceptors in bacteria is scarce. In *E. coli,* the Tar and Tsr chemoreceptors were found to be about ten times more abundant than Trg and Tap [14,15]. Cellular levels of 7 of the 8 chemoreceptors of *Sinorhizobium meliloti* could be quantified by immunoblot and were classified into high, low, and very low abundance receptors with an approximate cellular ratio of 300:30:1 [16]. The abundance of the 10 chemoreceptors in *Bacillus subtilis* has been quantified by immunoblot and differed by a factor of about 40 [17]. A number of studies demonstrate that chemoeffector affinity for its chemoreceptor determines the onset of response [18,19,20], whereas the cellular amount of chemoreceptors appears to be related to the magnitude of chemotaxis, since chemoreceptor overexpression increases chemotaxis [20,21,22,23].

We have assessed here the issue of sensor histidine kinase and chemoreceptor amounts using *P. aeruginosa* as a model organism. Since both receptor families can also be stimulated by the binding of signal-loaded solute binding proteins (SBPs) [24], this family has also been included in our study. *P. aeruginosa* is among the most feared human pathogens [25], able to infect almost all human tissues [26,27]. It is ubiquitously present in the environment [28], has a versatile lifestyle [29], and is a universal pathogen able to infect different animals and plants [30,31]. In 2019, *P. aeruginosa* infections caused the deaths of 559,000 people worldwide [25]. *P. aeruginosa* is also a central model organism for studying signal transduction by chemosensory pathways [9] and sensor histidine kinases [32]. Members of both receptor families are involved in mediating different virulence-related processes [9,32].

*P. aeruginosa* has five gene clusters that encode proteins of four different chemosensory pathways (Figure 1).

Whereas the Che pathway mediates chemotaxis, the Wsp (wrinkly spreader phenotype) pathway controls c-di-GMP levels, and the Chp (chemosensory pili) pathway is associated with type IV pili-based motility [9]. The function of the Che2 pathway is currently less clear, but it was found to tune the behavior of the Che pathway according to the levels of O_2_ sensed by McpB/Aer2 [33] and is required for full virulence [9,34,35]. *P. aeruginosa* PAO1, the model strain of this work, has 26 chemoreceptors (Figure 2), a number well above the bacterial average of 14 [36]. Bioinformatic predictions and experimental data indicate that 23 chemoreceptors stimulate the Che pathway, whereas the remaining three pathways are each stimulated by a single chemoreceptor (Figure 2) [8]. While the functional role of most bacterial chemoreceptors is unknown, the role of many *P. aeruginosa* chemoreceptors has been defined—information that is summarized in Table 1.

*P. aeruginosa* chemotaxis chemoreceptors have been identified that recognize and mediate chemoattraction to amino acids (PctA, PctB), γ-aminobutyrate (GABA) (PctC), nitrate (McpN), oxygen (Aer), histamine and polyamines (TlpQ), acetylcholine and other quaternary amines (PctD), α-ketoglutarate (McpK), malate and derivatives (CtpM), or inorganic phosphate (Pi) (CtpL, CtpH). In addition, *P. aeruginosa* PAO1 has 64 sensor histidine kinases, and their corresponding response regulators are known for 54 of them [32]. Several of these kinases interact with each other in complex multikinase networks [37]. Furthermore, PAO1 has 93 SBPs that were shown or predicted to bind a very wide range of different compounds [38,39]. Taken together, the wealth of information available on sensor protein function makes *P. aeruginosa* PAO1 a well-suited model to investigate chemoreceptor, sensor histidine kinase, and SBP abundance and sensing preferences. 

Among the environmental stimuli that induce *P. aeruginosa* virulence are low concentrations of Pi [40,41]. It has been shown that oral administration of Pi protected mice against infections caused by *P. aeruginosa* [42,43]. The resulting clinical strategy has been termed phosphate therapy, which corresponds to a promising alternative to antibiotics, particularly in the context that low Pi concentrations also induce virulence in a number of other bacterial and fungal pathogens [44,45,46,47,48]. Very important changes in global transcript levels have been observed when *P. aeruginosa* is grown in Pi deficient conditions (minimal medium supplemented with 0.2 mM Pi) as compared to growth in a Pi sufficient medium (minimal medium supplemented with 1 mM Pi) [40], and in a recent study we have reported that Pi depletion results in the upregulation of key metabolic pathways and virulence factors [49].

**Table 1 ijms-24-01363-t001:** *P. aeruginosa* PAO1 chemoreceptors for which information is available on function and/or signal sensed.

Locus Tag	Name	Effector (Binding Mode)	Function/Comment	References
PA0176	Aer2/McpB	O_2_ (direct)	Stimulates the Che2 pathway	[35,50,51,52,53]
PA0180	CttP/McpA	Chloroethylenes (unknown)	Chemotaxis	[54]
PA0411	PilJ	Phosphatidylethanolamine? PilA?	Stimulates the Chp pathway	[55]
PA1423	BdlA	Unknown	Involved in biofilm dispersion	[56,57,58]
PA1561	Aer/TlpC	O_2_ (unknown)	Aerotaxis	[51,59]
PA1930	McpS	Unknown	Modulates chemotaxis and chemoreceptor clustering	[60]
PA2561	CtpH	Inorganic phosphate (direct)	Chemotaxis	[61]
PA2573	-	Unknown	Involved in virulence	[62]
PA2652	CtpM	Malate, citramalate, methylsuccinate, bromosuccinate, citraconate (direct)	Chemotaxis	[63,64]
PA2654	TlpQ	Histamine, putrescine, cadaverine, spermidine, agmatine, ethylenediamine (direct), ethylene (unknown)	Chemotaxis	[65,66]
PA2788	McpN	Nitrate (direct)	Chemotaxis	[67]
PA3708	WspA	Growth on solid surfaces (unknown), ethanol (unknown)	Stimulates the Wsp pathway	[68,69]
PA4307	PctC	γ-aminobutyrate, histidine, proline (direct), histamine (unknown)	Chemotaxis	[65,70,71,72,73,74,75]
PA4309	PctA	17 amino acids (direct), histamine (unknown), chloroethylenes, chloroform (unknown)	Chemotaxis	[65,70,71,72,73,75]
PA4310	PctB	5 amino acids (direct)	Chemotaxis	[65,70,72,73,75]
PA4633	PctD	Acetylcholine and other quaternary amines (direct)	Chemotaxis	[20]
PA4844	CtpL	Inorganic phosphate (indirect, via PstS), chloroaniline, catechol (unknown)	Chemotaxis	[61,76]
PA5072	McpK	α-ketoglutarate (direct)	Chemotaxis	[77]

In this study, we report the protein amounts of *P. aeruginosa* chemoreceptors, chemosensory signaling proteins, sensor histidine kinases, and SBPs. Our data are complemented by chemotaxis assays and provide novel insights into sensing preferences and signaling mechanisms.

## 2. Results

*P. aeruginosa* PAO1 cells were grown in LB or minimal medium supplemented with 0.2 or 1 mM Pi [49]. No significant differences were observed in the growth kinetics between these three conditions [49]. At an OD_660_ of 0.6, cells were harvested, and proteins extracted and separated on SDS-PAGE gels. The SDS-PAGE gel used for the proteomics study is shown in Appendix A, revealing large changes in the proteome. Following digestion, peptides were identified by mass spectrometry. Proteomics-based absolute protein amounts are provided as log2iBAQT values that correspond to the logarithm in base 2 of the intensity Based Absolute Quantification (iBAQ).

### 2.1. Large Differences in Chemoreceptor Protein Amounts

This proteomics study has permitted us to determine the amount of approximately half of the proteins encoded in the genome of PAO1 [78], namely 2716, 2747, and 2733 proteins for growth in minimal medium with 0.2 mM Pi, 1 mM Pi, and LB, respectively (Appendix A).

Of the 26 chemoreceptors of PAO1, 20 could be quantified in at least 1 of the experimental conditions (Figure 3).

Among these receptors were PilJ, McpB/Aer2, and WspA, which are the chemoreceptors that stimulate the Chp, Che2, and Wsp pathways, respectively [8]. The remaining 17 chemoreceptors stimulate the Che chemotaxis pathway. As shown in Table 2, chemoreceptor protein amounts in a given growth condition vary greatly, up to 2420-fold considering all chemoreceptors and up to 592-fold considering only the chemotaxis chemoreceptors. 

However, the differences between the most and least abundant chemoreceptors are likely to be larger, since chemoreceptor amounts below the mass spectrometric detection limit are probably the reason for the failure to detect the remaining six chemoreceptors.

The most abundant chemoreceptor in LB was PilJ, followed by chemoreceptors that sense electron acceptors, namely oxygen (Aer and McpB/Aer2) [51,53] and nitrate (McpN) [67] (Figure 3). The amount of the third non-chemotaxis receptor, WspA, was about 60 and 12 times lower than the PilJ and McpB/Aer2 receptors, respectively. Significant protein levels were observed in LB for CtpM (chemoeffectors: malate and derivatives), PctB (primarily L-Gln), PctC (primarily GABA), and TlpQ (histamine and polyamines). Chemotaxis chemoreceptors with lower levels include PctA (multiple amino acids), PctD (quaternary amines), and McpK (α-ketoglutarate) (Figure 3). The chemoreceptors CtpH and CtpL were not detected in LB medium, in accordance with the fact that Pi downregulates *ctpH* and *ctpL* expression [40,61].

Our data also show that the amounts of a given chemoreceptor vary largely depending on the growth condition. A very important accumulation of McpB/Aer2 is observed under Pi deficiency (Figure 3). This agrees with transcriptomic studies [40] and with the notion that the Che2 pathway, including the McpB/Aer2 chemoreceptor, is required for virulence [34,35]. Two other receptors involved in virulence were increased under phosphate starvation, namely PA2573 [62] and the Pi-specific chemotaxis chemoreceptor CtpL [79] (Figure 3), a finding that also agrees with the transcriptomic study [40]. The importance of the growth medium for receptor levels is also illustrated by McpN and the Aer aerotaxis receptor, which were about 10 and 100 times more abundant in LB as compared to 0.2 mM Pi, respectively (Figure 3).

In subsequent experiments, we wanted to determine to which degree alterations in chemoreceptor amounts affect chemotaxis. To this end, we selected chemoreceptors that showed significant variation in different growth media and for which it was shown that the mutation of the corresponding gene abolished chemotaxis to a given chemoattractant. These receptors were PctC, PctD, and McpK, for which we have shown previously that a deletion or inactivation of the corresponding gene has suppressed chemotaxis to GABA [72], choline [20], or α-ketoglutarate [77], respectively. This implies that responses of the wild type strain to these compounds are entirely due to the action of these receptors. To establish links between chemoreceptor amounts and the magnitude of chemotaxis, we have cultured the wild type strain in the same way as in the proteomic study and conducted quantitative capillary chemotaxis assays to two different chemoeffector concentrations (Figure 4). Although there were some tendencies, we did not observe a statistically relevant correlation between chemoreceptor abundance and magnitude of chemotaxis (Figure 4).

### 2.2. Excess of Response Regulators over Sensor Histidine Kinases

Several pieces of evidence indicate that the cellular abundance of sensor histidine kinases is, in general, lower than that of chemoreceptors. As shown in Figure 3, 20 of the 26 chemoreceptors (i.e., 77%) could be detected in at least one growth condition with a mean log2iBAQ of 17.4 ± 2.6. In contrast, only 11 of the 64 sensor histidine kinases (i.e., 17%) could be detected with a mean log2iBAQ of 15.4 ± 2.3 (Figure 5). Very abundant in all three growth conditions were the NtrB sensor histidine kinase and its cognate response regulator NtrC, forming a two-component system important for the regulation of cellular nitrogen levels and nitrate assimilation [80].

Pi is sensed by the PhoRB two-component system that controls the expression of genes related to phosphate uptake and metabolism [81,82]. Although high levels of the sensor histidine kinase PhoB were identified in all three growth conditions, an important accumulation of PhoB and PhoR was observed under Pi deficiency. Also of note is the increased abundance of the uncharacterized two-component system PA2881/PA2882 under Pi depleted conditions. In contrast, the sensor histidine kinases PA4112 and NarX, with roles in biofilm formation and nitrate sensing, respectively [83], were not detected under Pi deficiency (Figure 5). The uncharacterized sensor histidine kinase PA1243 showed reduced levels at low Pi availability. The Gac system is among the best characterized multikinase networks that integrates stimuli through seven interconnected sensor histidine kinases, regulating many virulence-related traits such as biofilm formation, c-di-GMP levels, motility, or type III and VI secretion systems [83]. Three of the participating sensor histidine kinases, GacS, RetS, and BifS, could be detected at relatively stable levels in all three growth conditions (Figure 5).

Information on the cognate response regulators was available for 6 of the 11 sensor histidine kinases detected (Figure 5). In general, response regulators were significantly in excess over sensor histidine kinases. On average, considering the abundance of the 6 sensor histidine kinase/response regulator pairs under all growth conditions, there was about a 70-fold excess of response regulators over sensor histidine kinases (Figure 5), which is consistent with previously published data on other systems [84,85,86,87].

### 2.3. Amounts of Chemosensory Signaling Proteins

#### 2.3.1. Overall Abundance of Signaling Proteins from the Four Chemosensory Pathways

The amount of the chemosensory signaling proteins of the four pathways is shown in Figure 6, and the average log2iBAQ values of proteins that belong to the individual pathways are provided in Figure 7.

Most abundant are proteins of the Che chemotaxis pathway, and all seven chemosensory signaling proteins (Figure 1) could be detected at high levels (Figure 7 and Figure 8). Differences in the growth conditions caused few changes to the Che pathway protein levels (Figure 6). The amounts of the Che pathway signaling proteins are also consistent with the fact that 23 different chemoreceptors feed into this pathway [8]. The second most abundant pathway, for which seven chemosensory signaling proteins could be detected, was Chp (Figure 7 and Figure 8), which agrees with the observation that the corresponding chemoreceptor PilJ was the most abundant chemoreceptor (Figure 3). Proteins of the Che2 pathway were less abundant when grown in LB, but a considerable increase in protein amounts was observed under Pi starvation (Figure 7), which is consistent with transcriptomic analyses [40] and the role of the Che2 pathway in virulence [34,35]. Three proteins (CheW_3-1_, CheW_3-2_, and CheY_3_) of the Wsp pathway could be quantified in all three growth conditions (Figure 6). These three proteins are present in relatively low amounts and in very similar levels in the three growth conditions (Figure 7 and Figure 8). The low amount of the Wsp signaling proteins agrees with the modest abundance of the corresponding chemoreceptor, WspA (Figure 3).

#### 2.3.2. Differences in the Ratio of Signaling Proteins with the Individual CheA Homologs

##### Chemoreceptor:CheA Ratios

The CheA histidine kinase is the central protein of a chemosensory pathway, and our data permit us to gain information on the ratios of other signaling proteins with respect to CheA. Chemoreceptor:CheA ratios for the Che2 and Chp pathways that are stimulated by a single receptor ranged from 3.8:1 to 32:1 (Table 3). In contrast, the chemoreceptor:CheA ratios of the Che pathway, taking into account the sum of chemoreceptors, were much lower, between 0.25:1 to 0.4:1 (Table 3). Chemoreceptor:CheA ratios have been determined in *B. subtilis* [17], *E. coli* [88] and *S. meliloti* [16] to be 23:1, 6.8:1 and 23.5:1, respectively, indicating important differences among species. The elucidation of the physiological relevance of these different ratios is among the research needs of the field.

##### Coupling Protein:CheA Ratios

The Che pathway contains the canonical coupling protein CheW and the alternative coupling protein CheV, which corresponds to a fusion of CheW with a receiver domain. In general, the role of CheV homologs is less clear since at times it is redundant to CheW, whereas on other occasions it was shown to be essential for response generation [89]. In a chemosensory array, two CheW proteins interact with a CheA dimer [89]. As shown in Table 3, there is an excess of both CheW_1_ and CheV over CheA_1_, indicating ratios between 4.9 to 8.7 CheW/CheV coupling proteins per CheA_1_. In all three growth conditions, CheW_1_ was more abundant than CheV (Table 3). In contrast, the CheW_2_:CheA_2_ ratios were below 1 (Table 3), which again suggests a more intricate regulation of this pathway.

Genome analyses showed that bacteria frequently contain an elevated number of CheW and CheY homologs, suggesting that different homologs participate in pathway signaling [2,90]. There are two CheW homologs (CheW_4-1_ and CheW_4-2_) in the Chp pathway (Figure 1), making it a model to elucidate the functional relevance of multiple CheW homologs. In 1 mM Pi and LB, there was a 3.7- to 5.3-fold excess of the two CheW homologs over CheA_4_, whereas CheW: CheA_4_ ratios of 16.9 (CheW_4-1_: CheA_4_) and 26.9 (CheW_4-2_: CheA_4_) were determined in 0.2 mM Pi. Taken together, there were thus 8 to 44 CheW homologs (CheW_4-1_ and CheW_4-2_) per CheA_4_ in the different growth conditions.

##### Response Regulator:CheA Ratios

CheA transphosphorylates two response regulators, CheY and CheB. Previous studies of two-component systems [84,85,86,87] and our data (Figure 5) revealed a large excess of response regulators to sensor histidine kinases, indicating that this is a general property of two-component systems. However, in marked contrast were the ratios of CheY_1_ and CheB_1_ to CheA_1_ that were all below 1 (Table 3). Under all growth conditions, there appear to be about 0.5 to 1 response regulators (CheY_1_ + CheB_1_) per molecule of CheA_1_ (Table 3). This deviation from the general norm observed for two-component systems requires further investigation. CheY/CheB:CheA ratios of the Che2 pathway were similarly low, especially at Pi sufficient conditions (Table 3).

Previous studies have brought some insight into the function of the two CheY homologs of the Chp pathway, PilG and PilH, that are both phosphorylated by CheA_4_ [91,92] Whereas PilG interacts with FimV, the activator of the CyaB adenylate cyclase, causing alterations in the cAMP level, PilH was proposed to function as a phosphate sink [91,92]. Interestingly, there was a 39- to 170-fold excess of PilG/PilH over CheA_4_ that was reminiscent of response regulator:histidine kinase ratios in two-component systems. This ratio was particularly elevated in 0.2 mM Pi, where a ratio of 170 CheY homologs (PilH + PilG) per CheA_4_ molecule was observed. In contrast to the elevated PilH + PilG levels, there were only minor amounts of CheB_4_ (Table 3).

##### CheR:CheA Ratios

The antagonistic actions of the CheR methyltransferase and CheB methylesterase CheB control the receptor methylation state, enabling pathway adaptation [13,93]. Similar to CheB, the CheR:CheA ratios were in all cases below 1 (Table 3). Data thus indicate that low and balanced adaptor enzymes:CheA ratios are a more general feature of chemosensory pathways (Table 3).

### 2.4. Amount of Solute Binding Proteins

Sensor histidine kinases and chemoreceptors can be stimulated by the binding of signal-loaded SBPs [24]. There are a number of arguments suggesting that such sensing mechanisms are more frequent than currently believed [24]. Of the 93 SBPs encoded in the genome of *P. aeruginosa* PAO1, 62 (i.e., 67%) could be quantified in at least one growth condition (Figure 8). The ligands recognized by SBPs can be predicted by the TransportDB database [94], and we have shown recently by comparing the TransportDB output with experimental data that these predictions are highly precise [39].

Of the 28 SBPs predicted to bind proteinogenic amino acids [39], 23 (i.e., 82%) could be quantified in at least one condition (Figure 8). It is noteworthy that many amino acid sensing SBPs are among the highly abundant SBPs (labelled in red in Figure 8). Most abundant was BraC, the SBP of the high-affinity branched chain amino acid uptake system [95] that binds its cognate ligands with K_D_ values between 70 to 720 nM [38]. Among the very abundant amino acid SBPs are AatJ and AotJ, which specifically bind Asp/Glu or Arg, respectively [38]. The other very abundant SBP family was that of the polyamine binding proteins (Figure 8; labelled in green), for which 8 of the 13 family members could be quantified. Among the highly abundant polyamine binding SBPs are SpuD and SpuE, belonging to the major spermidine uptake transport system SpuDEFGH of *P. aeruginosa* [96]. There is evidence that this transport system, and in particular SpuE, participates in signal transduction since the deletion of *spuE* significantly reduced the expression of the type III secretion system, a major *P. aeruginosa* virulence factor [97]. Taken together, these data illustrate not only that *P. aeruginosa* has an elevated number of amino acid and polyamine binding SBPs, but also that these are very abundant in the cell, illustrating the central physiological relevance of amino acid and polyamine uptake and sensing.

Previous studies have shown that the phosphate-specific PstS SBP is the most abundant protein of *P. aeruginosa* when grown under Pi limitations [79,98]. This finding was confirmed by our analysis (Figure 8) since PstS was identified as the most abundant protein under Pi deficiency conditions with a log2iBAQ of 30.61. PstS is a model system for studying the multiple functions of SBPs. Next to providing Pi to the high-affinity phosphate transporter PstABC, the interaction of PstS with this transporter also modulates the activity of the PhoRB two-component system [9]. In addition, PstS interacts with the CtpL chemoreceptor, mediating chemotaxis to low Pi concentrations [79].

A particularly interesting system is the di/tripeptide ABC transporter DppBCDF that receives its substrates from five paralogous SBPs, termed DppA1 to DppA5, that differ in their ligand ranges [99]. Except for DppA2, the remaining four DppA homologs could be quantified in all three growth conditions (Figure 8). Among these, most abundant was DppA5, followed by DppA3, DppA1, and DppA4. Whereas DppA3 is a broad range dipeptide binder, DppA2 specifically binds tripeptides, whereas data suggest that DppA5 has a very narrow ligand range [38,99].

## 3. Discussion

Two-component systems and chemosensory signaling cascades are among the most important bacterial signal transduction systems. Whereas in a canonical two-component system one sensor protein controls the phosphorylation state of its cognate response regulator, in chemosensory pathways typically many chemoreceptors control the phosphorylation state of CheY. Chemoreceptor abundance is thus proposed to define the weight by which a given receptor contributes to defining the final response, as supported by different studies in which specific chemoreceptors were overexpressed [20,21,22,23]. Here we have used proteomics to analyze chemoreceptor protein amounts in *P. aeruginosa*—a model bacterium in the field of signal transduction [9,32]. A primary result of this study is the observation of large differences in the chemoreceptor amounts between growth conditions. For example, there was a more than 2400–fold difference between the least and most abundant chemoreceptors under Pi depletion (Table 2); a difference that is very likely to be considerably larger considering the difficulty of quantifying some low abundance receptors. The magnitude of these differences appears to be superior to those observed in other species, such as about a 10-fold difference between high and low abundance chemoreceptors in *E. coli* [14,15] or 300- and 40-fold differences between most- and least abundant chemoreceptors in *S. meliloti* [16] or *B. subtilis* [17], respectively.

The identification of environmental signals that define chemoreceptor abundance has been little explored and represents an important research need. Thus, considering the levels of chemotaxis receptors measured in LB rich medium, most abundant were Aer and McpN mediating chemotaxis to oxygen and nitrate, respectively, followed by CtpM, PctB, PctC, and TlpQ (Figure 3), suggesting sensing preferences for the corresponding chemoeffectors, namely malate, amino acids, and polyamines. In contrast, under Pi sufficient conditions in minimal medium, the chemotaxis receptors PctD and PctA for quaternary amines and amino acids, respectively, were the most abundant. Gaining access to nutrients appears to be the primary reason for chemotaxis [100]. Accordingly, the chemoreceptor CtpM (strong preference for malate [63,64]) was more abundant than McpK (α-ketoglutarate receptor [77]) under all growth conditions. This result coincided with experiments showing that PAO1 grew faster in the presence of D,L-malate as the sole carbon source as compared to α-ketoglutarate (Appendix A). Experiments were conducted to determine to what degree chemoreceptor abundance modulates the magnitude of chemotaxis. Although there were some tendencies between receptor abundance and chemotactic responses, no statistically relevant correlation was observed (Figure 4). This indicates that chemoreceptor abundance is only one of many factors that determine the magnitude of chemotaxis, such as ligand affinity for the receptor, efficiency of receptor activation, and signal transduction to the histidine kinase.

Chemoreceptor sensing preferences (Figure 3) also appear to be reflected in the amount of SBPs (Figure 8). Of the 93 SBPs of *P. aeruginosa,* 28, 13, and 10 were predicted to bind amino acids, polyamines, and quaternary amines, respectively [39]. Importantly, here we show not only that there is a high number of SBPs for amino acids and polyamine binding, but also that members of both protein families show an elevated cellular abundance. In fact, among the 20 most abundant SBPs identified in this study, 10 and 5 are predicted to bind amino acids and polyamines, respectively (Figure 8), suggesting an important role for amino acid and polyamine sensing in *P. aeruginosa*.

The chemosensory pathways Chp, Che2, and Wsp are each stimulated by a sole chemoreceptor, namely PilJ, McpB/Aer2, and WspA [8], which were detected under all growth conditions (Figure 3). PilJ is present at high levels in all three conditions, suggesting an important role in the corresponding Chp pathway. Given that proteins for this study were extracted from liquid cultures, this finding is somewhat surprising, considering that the function of the PilJ receptor is surface sensing and surface motility [101,102,103]. The highest levels of any chemoreceptor were for McpB/Aer2 under Pi scarceness, which agrees with transcriptomic studies [40] and its role in virulence [35]. WspA was detected at moderate levels under all conditions, which agrees with the low amounts of the corresponding pathway signaling proteins (Figure 6).

Several studies of two-component systems have reported that response regulators are in great excess over sensor histidine kinases [84,85,86,87], a notion that is also consistent with our data (Figure 5). The excess of response regulators over sensor histidine kinases has been shown to result in a robust output that is little dependent on alterations of the response regulator concentration [84,85,86,87]. Our data indicate that the Chp pathway operates by a similar mechanism, since there were ratios of 39 to 170 CheY homologs (PilH + PilG) per CheA_4_ (Table 3). This, however, is in marked contrast to the Che chemotaxis pathway for which CheY_1_ and CheB_1_ ratios to CheA_1_ were about 0.5 to 1 (Table 3), indicating that alterations in response regulator amounts may significantly modulate signaling. This potential disadvantage appears to be compensated by the fact that there were only minor variations in the amounts of the Che pathway signaling proteins under the different growth conditions (Figure 6).

Several studies have determined the chemoreceptor:CheA ratios in *E. coli.* Measurements were similar and comprised cellular ratios of 6.8:1 [88], 6:1 [104], and 6-9:1 [105]. However, in other species, very different chemoreceptor:CheA ratios were determined, such as 23:1 for *B. subtilis* [17] and 23.5:1 for *S. meliloti* [16]. The cryo-electron tomography analysis of chemosensory arrays from different species revealed a chemoreceptor:CheA ratio of 6:1, consistent with the measurements made in *E. coli* [106]. We were able to derive the PilJ:CheA_4_ and McpB/Aer2:CheA_2_ ratios in all three experimental conditions (Table 3). Five of these values were between 3.8:1 and 9:1 and are thus in the same range as the expected 6:1 ratio. However, a ratio of 32 McpB/Aer2 chemoreceptor:1 CheA_2_ was obtained in LB, a value similar to those measured in *B. subtilis* and *S. meliloti*. The reason for this differing stoichiometry appears to be the unusually low amount of CheA_2_ (Figure 6). We also show that chemoreceptor:CheA ratios vary significantly depending on growth conditions, with elevated ratios suggesting that the majority of chemoreceptors are not engaged in functional signaling complexes; further studies are necessary to understand the function and dynamics of chemosensory arrays. We were also able to quantify most but not all chemoreceptors that feed into the Che chemotaxis pathway. The failure to detect the remaining chemoreceptors is most likely due to their low abundance; as a consequence, their absence from the calculation of the chemoreceptor:CheA_1_ ratio would not significantly affect this value. The ratios of the sum of all chemotaxis receptors to CheA_1_ are between 0.25 and 0.4 chemoreceptors per CheA_1_ and are much lower than those involving CheA_2_ and CheA_4_ (Table 3), suggesting that there are CheA_1_ molecules that are not involved in contacts with chemoreceptors. Further studies are required to determine the functional reason for the different chemoreceptor:CheA ratios in the different pathways.

The cryo-electron tomography analysis of chemosensory arrays indicates a 1:1 CheA:CheW ratio [106]. However, the authors of this study also noted “empty hexagons” in the array lattice that correspond to a 1:2 CheA:CheW ratio [106]. However, a CheW:CheA_4_ ratio of 44:1 (CheW_4-2_ + CheW_4-1_) (Table 3) suggests that the large majority of CheW molecules are not engaged in signaling complexes in the Chp pathway. The functional relevance of this observation, in context with the much lower CheW:CheA ratios of the other pathways, still needs to be determined.

The signaling output of the four chemosensory pathways is different [3], but it remains to be established to what degree their molecular mechanisms differ. Important findings of our study include that: (i) the amount of signaling proteins differs significantly among pathways (Figure 6 and Figure 8); (ii) the ratios of chemoreceptors CheY and CheW to CheA vary significantly among pathways (Table 3); and (iii) the ratios of CheR and CheB to CheA are in a very similar range (Table 3). The differences observed are likely the reflection of differences in the mode of action of these pathways.

SBPs stimulate chemoreceptors and sensor histidine kinases by binding to the extracytoplasmic sensor domain [24]. There was a great excess of SBPs as compared to chemoreceptors and sensor histidine kinases. For example, levels of the most abundant SBP, BraC, were 18 and 207 times higher than those of the most abundant chemoreceptor, PilJ, and sensor histidine kinase, NtrB, respectively. So far, the only characterized example of SBP mediated chemoreceptor stimulation in *P. aeruginosa* is the activation of the CtpL chemoreceptor by PstS binding [79]. Both proteins were quantified in 0.2 and 1 mM Pi, and PstS/CtpL ratios of 3848:1 and 4240:1 were determined, respectively, indicating an enormous excess of PstS over CtpL. The excess of SBPs over chemoreceptors and sensor kinases is most likely due to the fact that the primary function of SBPs is providing solutes to transporters.

Chemosensory pathways are considered to be the most complex signal transduction systems as evidenced by bacteria with 90 different chemoreceptors [12], species that contain up to 8 different pathways [7], or pathways that contain alternative auxiliary signaling proteins [3]. The complexity in the protein composition of these pathways is reflected by complex molecular mechanisms. We show here the usefulness of proteomics to obtain information on the amounts of signal transduction proteins, giving in turn useful insight into sensing preferences and mechanisms. Our study may serve as a reference for similarly analyzing other bacteria.

## 4. Materials and Methods

### 4.1. Proteomics

The experimental protocol of sample preparation, proteomics studies, and data analyses have been published in [49]. Briefly, proteins were extracted in 6 M urea, 2 M thiourea, 50 mM Tris/HCl, pH 7.5, and 25 µg of protein were separated by SDS-PAGE. Gel lanes were fractionated into 10 gel pieces, and proteins were in-gel digested following the protocol published previously [107]. Peptides were subjected to LC-MS/MS analyses on a LTQ Orbitrap VelosPro instrument (ThermoFisher Scientific, Waltham, MA, USA) coupled to an EASY-nLC II liquid chromatography system. Peptides were loaded on a self-packed analytical column (OD 360 µm, ID 100 µm, length 20 cm) filled with 3-µm diameter C18 particles (Maisch, Ammerbuch-Entringen, Germany) and eluted by a binary nonlinear gradient of 5% to 99% acetonitrile in 0.1% (*vol*/*vol*) acetic acid over 87 min with a flow rate of 300 nL/min. For MS analysis, a full scan in the Orbitrap with a resolution of 30,000 was followed by collision-induced dissociation (CID) of the 20 most abundant precursor ions. Database searches against a database of *P. aeruginosa* PAO1 downloaded from the *Pseudomonas* Genome DB (https://www.pseudomonas.com) on 6 October 2021 (5587 entries) were performed using MaxQuant (version 1.6.17.0), combining the 10 MS-runs that correspond to the same sample (gel lanes) into one search output. The false discovery rates (FDRs) of protein and peptide spectrum match (PSM) levels were set to 0.01, and 2 unique peptides were required for protein identification. The iBAQ (intensity Based Absolute Quantification) data [108] were exported from MaxQuant (version 1.6.17.0) as a proxy for protein abundance. Briefly, iBAQ uses the sum of peak intensities of all peptides matching a specific protein, which is divided by the number of theoretically observable peptides. Hence, obtained protein amounts can be compared between different samples but also within a given sample.

### 4.2. Quantitative Capillary Chemotaxis Assays

Overnight cultures of *P. aeruginosa* PAO1 were grown at 30 °C with orbital shaking (200 rpm) in LB (5 g/L yeast extract, 10 g/L bactotryptone, 5 g/L NaCl) or minimal medium (MM; 0.1 M HEPES, 7 mM (NH_4_)_2_SO_4_, 1 mM MgSO_4_, 6 mg/L Fe-citrate, trace elements [109], pH 7.0) containing 20 mM succinate as carbon source and 0.1–50 mM K_2_HPO_4_/KH_2_PO_4_. Overnight cultures were washed twice (centrifugation at 1667× *g* for 5 min followed by resuspension in 10 mM HEPES, pH 7.0) and then used to inoculate fresh LB or MM to an initial OD_660_ of 0.05. Cells were cultured at 37 °C until an OD_660_ of 0.4–0.5. Subsequently, cells were washed twice and gently resuspended in 1 mL of 10 mM HEPES, pH 7.0. Cells were then diluted in 10 mM HEPES, pH 7.0 to an OD_660_ of 0.1, and 230 μL aliquots were placed into wells of 96-well microtiter plates. One-μL capillaries (Microcaps, Drummond Scientific, Broomall, PA, USA) were heat-sealed at one end and filled with chemoeffector solution prepared in 10 mM HEPES, pH 7.0. Subsequently, the capillaries were washed with sterile water, immersed into the bacterial suspensions at their open end, and incubated for 30 min. Capillaries were then removed, rinsed with sterile water, and emptied into 1 mL of 10 mM HEPES, pH 7.0. Serial dilutions were plated on M9 minimal medium [109] plates supplemented with 20 mM glucose, prior to an incubation at 30 °C overnight. Colonies were counted, and the data were corrected with the number of cells that swam into the 10 mM HEPES buffer (pH 7.0) containing capillaries. Data are the means and standard deviations of three biological replicates, conducted in quadruplicate.

### 4.3. Growth Experiments

PAO1 was grown overnight in M9 minimal medium containing 10 mM glucose. Overnight cultures were washed twice and then diluted to an OD_600_ of 0.02 in M9 medium supplemented with either 10 mM D,L-malate or α-ketoglutarate as the sole carbon source. Then, 100 µL of these cultures were transferred into a microwell plate and growth at 37 °C was followed on a Bioscreen microbiological growth analyzer (Oy Growth Curves Ab Ltd., Helsinki, Finland).

## Figures and Tables

**Figure 1 ijms-24-01363-f001:**
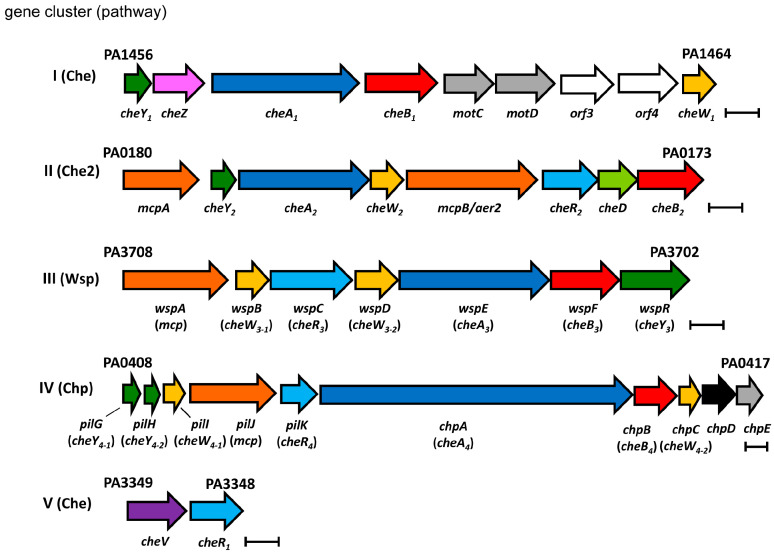
Schematic representation of the gene clusters of *P. aeruginosa* PAO1 that encode chemosensory signaling proteins. Che: chemotaxis; Wsp: wrinkly spreader phenotype; Chp: chemosensory pili. The Che2 pathway is of unknown function. Bars, 0.5 kbp.

**Figure 2 ijms-24-01363-f002:**
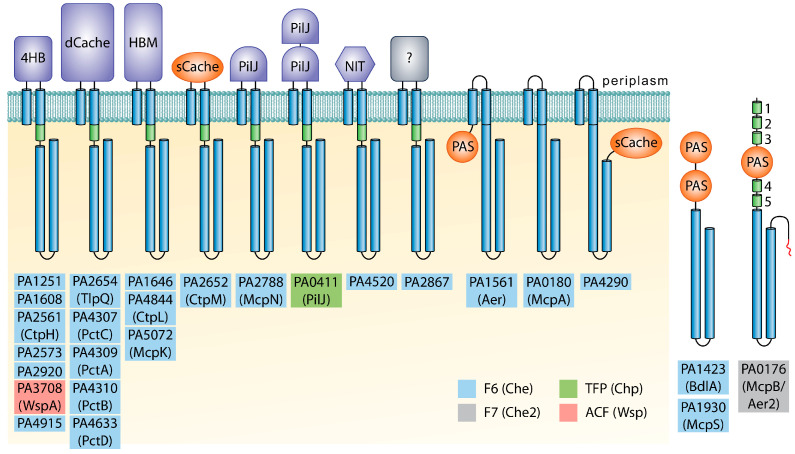
The chemoreceptor repertoire of *P. aeruginosa* PAO1. Ligand binding domains with parallel helix or α/β folds are shown in purple and orange, respectively. HAMP (histidine kinases, adenyl cyclases, methyl-accepting proteins, and phosphatases) domains are represented as green cylinders. 4HB, four helix bundle; Cache, calcium channels and chemotaxis receptors; HBM, helical bimodular; PilJ, N-terminal domain of type IV pili chemoreceptor; NIT, nitrate and nitrite sensing; PAS, Per–Arnt–Sim. The shading of the chemoreceptor name indicates the pathway these receptors were proposed to stimulate according to [8]. Figure reproduced from [9].

**Figure 3 ijms-24-01363-f003:**
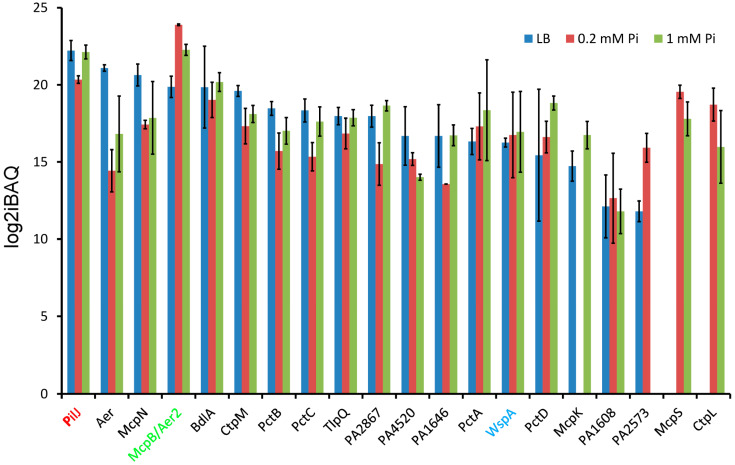
Chemoreceptor protein amounts in *P. aeruginosa* PAO1 grown in LB medium or minimal medium with 0.2 mM and 1 mM Pi. Chemoreceptors were ordered according to their protein amount in LB. Chemotaxis chemoreceptors are shown in black, whereas the chemoreceptors that stimulate the Chp, Che2, and Wsp pathways are shown in red, green, and blue, respectively. Error bars indicate standard deviations from the mean intensity Based Absolute Quantification (iBAQ) index.

**Figure 4 ijms-24-01363-f004:**
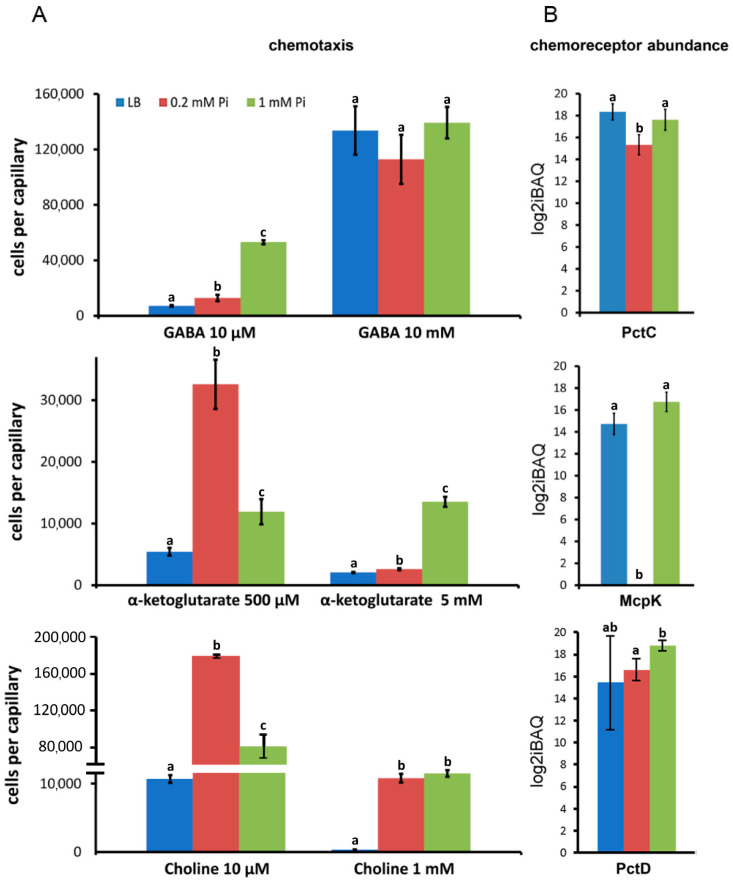
Relationship between the magnitude of chemotaxis and the cellular abundance of the corresponding chemoreceptors. (**A**) Quantitative capillary chemotaxis assays of *P. aeruginosa* PAO1 to GABA, α-ketoglutarate, and choline. Cells were grown the same way as for the proteomics analyses [49]. Responses have been corrected with the number of cells that swam in buffer-containing capillaries, namely 7350 ± 3850 (GABA), 6400 ± 2667 (α-ketoglutarate), and 7500 ± 3500 (choline). (**B**) Cellular protein amounts of the PctC, McpK, and PctD chemoreceptors in *P. aeruginosa* PAO1 grown under the conditions used for the chemotaxis assays. Differences between bars with the same letter are not statistically significant (*p*-value < 0.05; by Student’s *t*-test).

**Figure 5 ijms-24-01363-f005:**
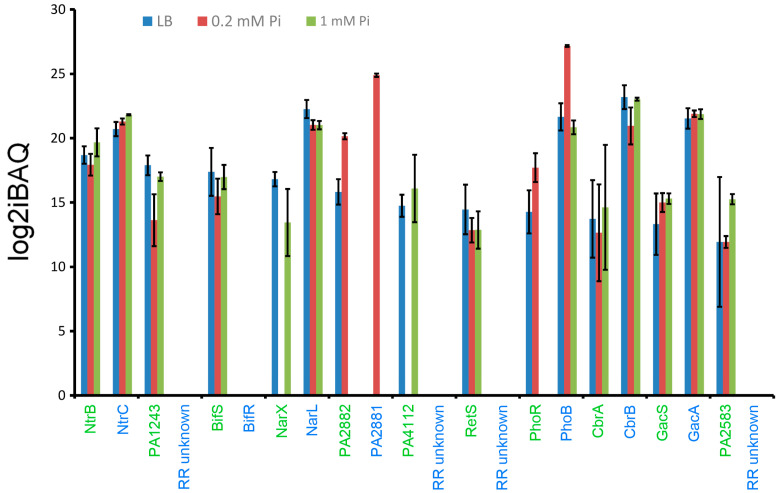
Amounts of sensor histidine kinases and response regulators of *P. aeruginosa* PAO1 grown in LB medium or minimal medium with 0.2 mM and 1 mM Pi. Data are ordered according to the protein amounts of sensor histidine kinases in LB. Sensor histidine kinases are annotated in green and their cognate response regulators in blue. Error bars indicate standard deviations from the mean intensity Based Absolute Quantification (iBAQ) index.

**Figure 6 ijms-24-01363-f006:**
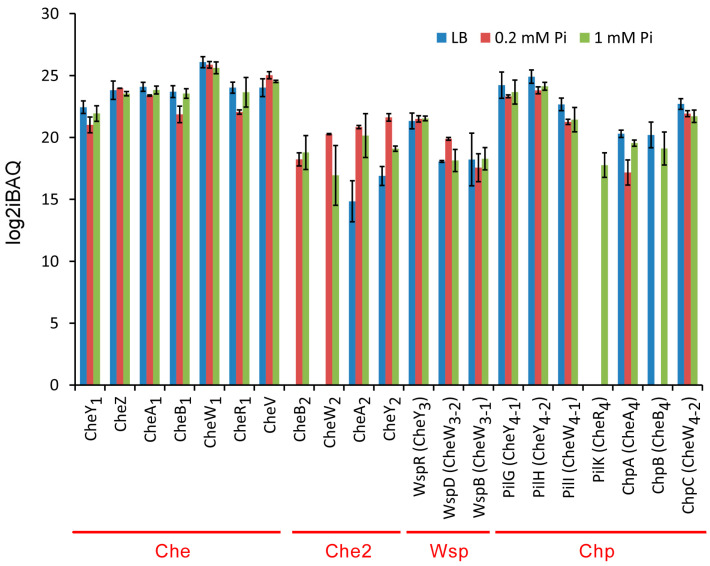
Absolute amounts of the chemosensory signaling proteins of the 4 chemosensory pathways of *P. aeruginosa* PAO1 grown in LB medium or minimal medium with 0.2 mM and 1 mM Pi. The corresponding pathways are indicated in red. Error bars represent standard deviations from the mean intensity Based Absolute Quantification (iBAQ) index.

**Figure 7 ijms-24-01363-f007:**
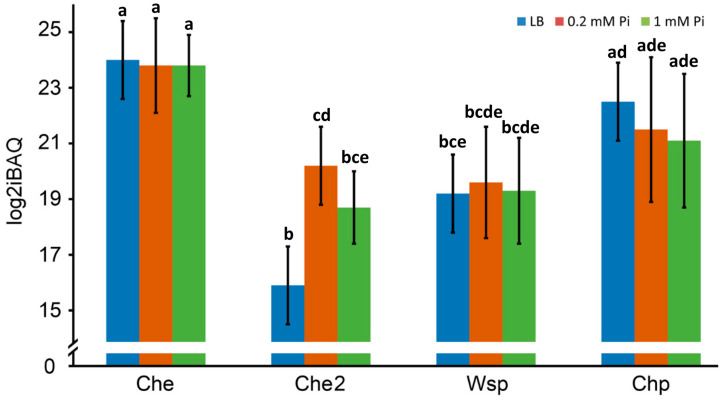
Mean abundance of the signaling proteins that form the 4 chemosensory pathways in *P. aeruginosa* PAO1. The values correspond to the sum of each of the proteins that were quantified under each experimental condition. Differences between bars with the same letter are not statistically significant (*p*-value < 0.05; by Student’s *t*-test).

**Figure 8 ijms-24-01363-f008:**
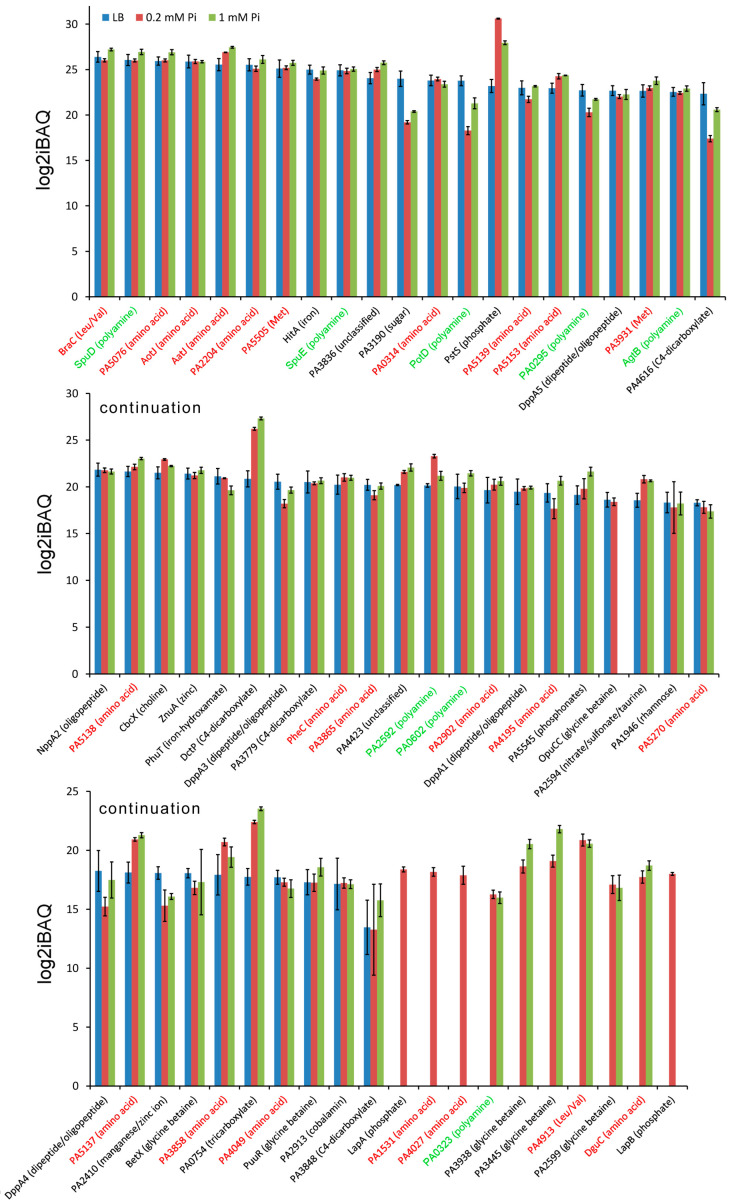
Relative abundance of solute binding proteins in *P. aeruginosa* PAO1 grown in LB medium or minimal medium with 0.2 mM and 1 mM Pi. Proteins were ordered according to their protein amount in LB. The ligands for these proteins were predicted by Transport DB [94]. The experimental determination of ligand profiles for *P. aeruginosa* PAO1 solute binding proteins largely confirms these predictions [38]. Proteins predicted to bind amino acids or polyamines are shown in red and green, respectively. Error bars indicate standard deviations from the mean intensity Based Absolute Quantification (iBAQ) index.

**Table 2 ijms-24-01363-t002:** Most and least abundant chemoreceptors and chemotaxis chemoreceptors of *P. aeruginosa* PAO1 grown in LB, minimal medium + 1 mM Pi, and minimal medium + 0.2 mM Pi.

Most/Least Abundant Chemoreceptors	iBAQ Value	Ligand/Comment	Ratio Most/ Least Abundant
Total chemoreceptors in LB			
Most: PA0411 (PilJ)	4,885,246	Stimulates Chp pathway	1370
Least: PA2573	3565	Unknown/involved in virulence	
Total chemoreceptors in minimal medium + 1 mM Pi			
Most: PA0176 (McpB/Aer2)	5,022,589	Oxygen/stimulates Che2 pathway	1408
Least: PA1608	3566	Unknown	
Total chemoreceptors in minimal medium + 0.2 mM Pi			
Most: PA0176 (McpB/Aer2)	15,545,568	Oxygen/stimulates Che2 pathway	2420
Least: PA1608	6.422	Unknown	
Chemotaxis chemoreceptors in LB			
Most: PA1561 (Aer)	2,111,738	Oxygen/aerotaxis	592
Least: PA2573	3565	Unknown/involved in virulence	
Chemotaxis chemoreceptors in minimal medium + 1 mM Pi			
Most: PA1423 (BdlA)	1,185,289	Unknown/involved in biofilm dispersion	332
Least: PA1608	3566	Unknown	
Chemotaxis chemoreceptors in minimal medium + 0.2 mM Pi			
Most: PA1930	763,954	Unknown	119
Least: PA1608	6422	Unknown	

**Table 3 ijms-24-01363-t003:** Ratios of chemoreceptors and signaling proteins to CheA orthologues in different growth conditions.

Signaling Protein	MM + 0.2 mM Pi	MM + 1 mM Pi	LB
**Chemoreceptors**			
**Che receptors vs. CheA_1_ ^a^**	0.25	0.28	0.4
**McpB/Aer2 vs. CheA_2_**	8.2	4.3	32
**PilJ vs. CheA_4_**	9.0	6.0	3.8
**Coupling proteins**			
**CheW_1_ vs. CheA_1_**	5.6	3.4	3.9
**CheV vs. CheA_1_**	3.1	1.6	1.0
**CheW_2_ vs. CheA_2_**	0.67	0.1	- ^b^
**CheW_4-1_ vs. CheA_4_**	16.9	3.7	5.1
**CheW_4-2_ vs. CheA_4_**	26.9	4.5	5.3
**Response regulator homologs**			
**CheY_1_ vs. CheA_1_**	0.19	0.26	0.32
**CheY_2_ vs. CheA_2_**	1.7	0.47	4.1
**PilG (CheY_4-1_) vs. CheA_4_**	99	24	24
**PilH (CheY_4-2_) vs. CheA_4_**	71	17	15
**CheB_1_ vs. CheA_1_**	0.35	0.81	0.76
**CheB_2_ vs. CheA_2_**	0.16	0.38	- ^b^
**CheB_4_ vs. CheA_4_**	- ^b^	0.74	0.93
**CheR**			
**CheR_1_ vs. CheA_1_**	0.39	0.87	0.95
**CheR_4_ vs. CheA_4_**	- ^b^	0.29	- ^b^

^a^ Sum of chemoreceptors that were predicted to feed into the Che pathway (in black in Figure 3). ^b^ Proteins not detected in the proteomic analysis under this growth condition.

## Data Availability

The MS data have been deposited to the ProteomeXchange Consortium via the PRIDE partner repository [110] with the dataset identifier PXD033250 (http://www.ebi.ac.uk/pride/archive/projects/PXD033250, accessed on 14 April 2022).

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
