# Peer review of "The Cellular Abundance of Chemoreceptors, Chemosensory Signaling Proteins, Sensor Histidine Kinases, and Solute Binding Proteins of Pseudomonas aeruginosa Provides Insight into Sensory Preferences and Signaling Mechanisms"

_ijms, 2023, doi:10.3390/ijms24021363_

Round 1

Reviewer 1 Report

The manuscript by Matilla et al. provides information about the abundance of chemosensory proteins in Pseudomonas aeruginosa using a relative abundance iBAQ mass spectrometry approach. P. aeruginosa possesses four chemosensory systems that control different cellular functions. Information about the relative abundance of their components under various phosphate concentrations will provide a useful tool for further studies. The manuscript is well written and experiments are designed, although some conclusions were overestimated. This work will add valuable knowledge to the field of bacterial chemosensing and signal transduction.

Major changes:

1.      Line 148: iBAQ for quantitation probably does give a reasonable measure of relative abundance, but the authors should define in greater detail what iBAC is, and how absolute numbers are being obtained with this method. Original references should be provided. E. g. Bing He, Jian Shi, Xinwen Wang, Hui Jiang, Hao-Jie Zhu, J. “Label-free absolute protein quantification with data-independent acquisition” Proteomics. 2019 May 30; 200:51-59. doi: 10.1016/j.jprot.2019.03.005. Epub 2019 Mar 14, and references 4, 5, and 6 within.

The software that was used to generate the iBAC numbers should be explicitly stated and referenced.

If the authors analyzed data from a single gel lane as 10 runs and then combined that data for one search this should be specifically mentioned.

2.      Fig. 4: This reviewer disagrees with the statement about the correlation between strength of chemotaxis response on chemoreceptor abundance. No statistical analyses have been provided to support this statement. (i) The error bars for the chemotaxis response to 10 mM GABA overlap; (ii) the response to 5 mM alpha-ketoglutarate is the same in LB and 0.2 mM Pi although there is a great difference between McpK abundance. There is only a small increase of McpK abundance at 1 mM Pi (not clearly significant) but a significant difference in chemotaxis response to alpha-ketoglutarate; (iii) similar trends are seen for 1 mM choline and PctD. This part of the study should be adjusted accordingly or the statements omitted completely.

3.      Line 412 and elsewhere: (as stated under major changes point 2): The abundance of a chemoreceptor alone is not necessarily defining its importance in a final chemotaxis response. The affinity of the chemotaxis ligand to the receptor, its subsequent activation, and signal transduction to the chemoreceptor kinase are at least three other contributing factors. Three studies have been referenced that support this statement but there are many others that do not. In fact, data provided in this study (Fig. 4) are also not in line with this conclusion. The relationship between chemoreceptor abundance and chemotaxis response should be more carefully stated and discussed.

4.      Line 513: This reviewer is not surprised that the ratio between SBPs and chemoreceptors is large. The primary function of SBPs is not chemotaxis but the binding of solutes and presenting them to their transporters. This statement should be more carefully stated and discussed.

Minor changes:

1.      Line 107: Suggestion ‘While the functional role of many bacterial chemoreceptors is unknown, the role of the majority of P. aeruginosa chemoreceptors has been defined.

2.      Line 123: Suggestion ‘and their corresponding response regulators are known for 54 of them.’

3.      Fig. 4: Scientific notation should be used on y-axis.

4.      Line 237: Fig. 5 legend: Define blue and green color coding on x-axis.

5.      Line 263: Paragraph 2.3.1 should be omitted; it does not contribute to the main line of the research presented in this manuscript.

6.      Line 409: Should read ‘signal transduction systems.’

Reviewer 2 Report

This manuscript by Matilla and colleagues combines proteomics with chemotaxis assays. Protein abundance in Pseudomonas aeruginosa was compared between phosphate-depleted and phosphate-sufficient media, with a focus on chemosensory proteins and two-component systems (histidine kinases and resonse regulators). The introduction section provides a comprehensive and informative review of the chemosensory proteins and two-component systems of P. aeruginosa and it was a pleasure to read. 

At the beginning of the Results section, I would have found it helpful to read a sentence or two outlining the general experimental procedure (rather than having to go to the Methods section to work it out). Adding it would be useful for the uninitiated reader. The fact that this much data can be obtained from just a few polyacylamide gels is impressive. I was left wondering though about the limitations of the methods used, and whether the methods could skew the results, thus affecting calculated protein ratios? Perhaps the authors could comment on this, including possibilites other then low protein levels for not detecting some proteins, such as native expression at different ODs. According to the paper referred to with the complete methodological details, cell samples were sacrificed at an OD660nm of 0.6. It is surprising that Che2 proteins were detected to the levels that they were, given that Che2 is expressed in stationary phase. 

Please note that Figure 9 was missing from the version of the MS that I read. 

Finally, I have a number of English language corrections/suggestions below. 

Minor corrections and comments

1.    Line 14 – I think you mean “traits” rather then “treats”

2.    Line 16 – The hyphen is unnecessary unless there is also one between “sufficient” and “media”

3.    Line 34 - Remove “and” before “protein kinases”

4.    Line 42 – For improved understanding, and to fit with the text on lines 46-49, remove the “s” from “regulators”.

5.    Line 61 – Please rework “plays key role in the system adaptation”. Perhaps “plays a key role in the adaptation of the system…” or something similar. 

6.    Line 63 – Add “s” to “system” and add a comma afterwards

7.    Line 67 – Change “in defining” to “of defining” and change “define” to a different word, e.g., determine, delineate, control. 

8.    Line 68 – Please end the sentence after “chemoeffector mixtures” so that it is not a run-on sentence. Delete the remaining text or create a new sentence with the remaining information.

9.    Line 83 – Remove “d” from “increased”

10. Line 85 -  Add “the” or “a” before “model”

11. Line 87 – Change “into” to “in”

12. Line 87 – P. aeruginosa is an important opportunistic pathogen, but I don’t agree that is it is “among the most feared human pathogens”. Please rework. 

13. Lines 102-103 – One known output of the Che2 pathway is tuning the behavior of the Che system according to the level of O2 sensed by Aer2 (through CheD, Orillard and Watts, Mol Micro, 2021).

14. Fig. 2 – The blue LBDs appear purple (on the screen and printed out) compared to the blue KCDs. 

15. Table 1 – For Aer2, remove the period after pathway in column 4. Also, ref 34 doesn’t really belong here. I am not sure if NO, CO, or cyanide belong here – they bind PAS-heme, but are not likely to be physiologically relevant. For McpS, left justify the text in the fourth column. 

16. Line 131 – Spell out “PA”

17. Line 136 – Add “Pi” after “0.2 mM”

18. Line 142-145 – These lines should be a part of the section before rather than a new paragraph. 

19. Line 143-145 – Change to “Our data are complemented by chemotaxis assays and provide novel insight…….” 

20. Line 152 – Add “us” or something similar between “permitted” and “to”

21. Line 179 – Remove “that is”

22. Line 181 – Change “is” to “was”

23. Line 208 – Remove the “s” from “ways”

24. Lines 214-216 – The strongest response to a-ketoglutarate is with 0.2 mM Pi and 500 mM inducer (which is not explicity discussed). Several sentences back you mention “higher chemoeffector concentrations”, which is what the sentence on lines 214-216 is referring to, but it would help to mention it in the sentence since it is several lines removed. I was looking at the graph in the context of the sentence and found it confusing.  

25. Figure 4 – Please provide statistical analysis of the bars within each inducer concentration. For instance, is there any significant difference in the bars for GABA 10 mM?

26. Figure 4 – Could differences in the constituents of LB vs minimal media affect the behavior of P. aerguinosa independent of differences in chemoreceptor concentrations? e.g., effects on cell size, swim speed, etc? 

27. Lines 224-226 – This statement is not needed to interpret the graphs and is mentioned in the text, so it can be deleted from the figure legend.  

28. Line 299 – Change “very little” to “less”

29. Line 311 – Add “us” after “permit”

30. Line 328 – Change the first “in” to “on”

31. Line 334 – Swap “contain frequently” with “frequently contain”

32. Line 365 – Change “Similarly” to “Similar”

33. Line 371 – Change “that” to “than”

34..Line 377 – Figure 9 is missing from the MS!

35. Line 393 – Italicize “P. aeruginosa”

36. Line 405 – Swap “binds specifically” with “specifically binds”

37. Lines 414-415 – Change to “Here we have used proteomics…”

38. Line 420 – Change “to quantify” to “in quantifying”

39. Line 452 – Swap “stimulated each” with “each stimulated”

40. Line 460 – “but little variable”makes no sense. Please rework. 

41. Paragraph beginning on Line 474 – What were the growth conditions for the cryo-EM studies, or any of the other mentioned studies, and is it relveant here?

42. Lines 486-498 – This jives well with fluoresence microscopy experiemnts, e.g., on Aer2 (Anaya et sl., 2002 and Guvener et al., 2006) showing that only a proportion of P. aeruginosa cells have polar Che2 clusters. 

43. Line 493 – Remove “in” before “between”

44. Line 516 – Change “The so far only” to “So far the only”

45. Line 522 – Change “evidence” to “evidenced”

46. Line 524 – Add “the” before “protein” and change “in” to “by”

47. Line 583 – Change “quadrupled” to “quadruplicate”

Reviewer 3 Report

Chemosensory pathways and two-component systems are very complex signal transduction systems. In model bacteria such as E.coli or Salmonella species, the number of components involved in the system are small, and thus the scheme is simple. However, in some other species, the number of components are very large, and pathways are not one.

This paper challenges the complex interaction among components of chemosensory pathways using proteomics and measured the relative abundance of components. The results are not simple to understand the relationship among components. However, this is a laborious work and should be published.

Round 2

Reviewer 1 Report

The authors addressed all of this reviewer's concerns.